| Open Peer Review | Immunology | Methods and Protocols

# Inter-laboratory harmonization of microsphere immunoassays for SARS-CoV-2 antibody detection in contrived dried blood spots and oral fluids

Kate L. DeRosa,[1] Nora Pisanic,[2] Kate Kruczynski,[2] Christopher D. Heaney,[2,3,4] Linda M. Styer,[1,5] Nicholas J. Mantis[1]

**ABSTRACT**    Dried blood spots (DBS) and oral fluids (OF) are easily attainable biospecimen types that enabled population-scale antibody monitoring for SARS-CoV-2 exposure and vaccination. However, the degree to which the two different biospecimen types can be used interchangeably remains unclear. To address this question, we generated contrived DBS (cDBS) and contrived OF (cOF) from serum panels of SARS-CoV-2-infected, vaccinated, and uninfected individuals. The contrived samples were evaluated using SARS-CoV-2 multiplexed microsphere immunoassays (MIAs) at two different institutions. Intra-laboratory tests revealed near-perfect agreement between cDBS and cOF for N and S antigens, as evidenced by $\kappa = 0.97–1$ and 98%–100% agreement. Interlaboratory comparisons were equally robust for both N ($\kappa = 0.94–0.96$; 97.5%–98 % agreement) and S ($\kappa = 0.98–1.0$; 99.0%–100%). Furthermore, assays were transferred between labs, including methods and reagents, and a subset of cDBS and cOF samples ($n = 52$) were tested. Qualitative concordance remained high ($\kappa = 0.94–1.0$; 97.5%–100% agreement), confirming that the integrity of the assays is retained upon transfer. In summary, our results provide evidence that DBS and OF can be used interchangeably across laboratories and institutions for the qualitative assessment of SARS-CoV-2 antibody determinations.

**IMPORTANCE**    The establishment of common research and clinical laboratory methodologies for the assessment of SARS-CoV-2 antibody levels in different biospecimen types is of paramount importance in estimating immunity to COVID-19 infections in communities and populations. In this report, we evaluated multiplex immunoassay protocols to enable comparisons between two readily attainable biospecimen types, namely, dried blood spots (DBS) and oral fluids (OF), which are employed in population-scale antibody monitoring of SARS-CoV-2 antibody titers and virus-neutralizing potential.

**KEYWORDS**    serology, COVID, antibody, mucosal, saliva, dried blood spots

COVID-19 continues to persist globally due to the extremely high transmissibility rates of SARS-CoV-2 and its ever-evolving variants of concern (VoC). From a public health standpoint, real-time estimates of the vulnerability of distinct cohorts across all age groups to circulating SARS-CoV-2 variants are of paramount importance when making decisions about next-generation vaccine implementation. While serological screening can be conducted as a routine part of health-care visits, such an approach will have a limited scope and be biased toward specific subpopulations. An alternative is to perform serosurveys in the field using low-cost methods that are conducive to self-collection and biospecimen retrieval and then send samples to one or more centralized testing sites for analysis of antibody and functionality against SARS-CoV-2 VoCs.

**Peer Reviewer** Dean D. Erdman, Centers for Disease Control and Prevention, Atlanta, Georgia, USA

Address correspondence to Christopher D. Heaney, cheaney1@jhu.edu, Linda M. Styer, linda.styer@health.ny.gov, or Nicholas J. Mantis, Nicholas.Mantis@health.ny.gov.

The authors declare no conflict of interest.

See the funding table on p. 8.

Dried blood spots (DBS) and oral fluids (OF) are two biospecimen types amenable to self-collection at a cohort, community, and even population scale (1–6). DBS are not only amenable to self-collection with preassembled kits and simple instructions, but the resulting spots are stable on Whatman filter paper for long periods of time under refrigeration (4–8°C) or frozen (up to −20°C) in the presence of desiccants (4, 7). Similarly, OF represent an easily accessible biospecimen type to interrogate both systemic and mucosal immune responses even among children (3). Thus, DBS and OF are complementary in the sense that they are convenient to collect and amenable for population-wide surveys. However, the degree to which the saliva and DBS are interchangeable in terms of SARS-CoV-2 antibody responses has not been established. In this study, we performed a comprehensive side-by-side comparison of paired contrived dried blood spots (cDBS) and contrived oral fluids (cOF) specimens using two SARS-CoV-2 microsphere immunoassays (MIAs) to evaluate the comparative strengths and weaknesses of each sample type.

## MATERIALS AND METHODS

### Microsphere immunoassays (MIAs)

In Lab A, antigens and controls were covalently coupled to magnetic microparticles (hereafter beads, Luminex xMAP) using carbodiimide coupling chemistry with sulfo-N-hydroxysuccinimide (sulfo-NHS) as described previously (5 µg protein/$1 \times 10^6$ beads, see Table S1 for list of antigens). The assay included a control bead indicating nonspecific binding of sample to beads. After activation with the crosslinker, assay buffer (PBS-TBN, Luminex, TX) containing 1 mg/mL BSA was added to the bead set. Each well contained 1,000 beads per set in the assay buffer. Detailed assay conditions are provided in Table 1. Assay plates included a blank (assay buffer) on each plate. Plates were read on a MagPix instrument. The blank was subtracted to calculate the net median fluorescence intensity (MFI), followed by subtracting the within-sample background (BSA bead net MFI). Cutoffs were defined based on the receiver-operator characteristic (ROC) area under the curve (AUC) for each antigen. Samples above the cutoff of nucleocapsid (GenScript N) and spike (Mt. Sinai S) were classified as "prior infection." Samples negative for N but positive for S were classified as "prior vaccination." For cOF, the RBD result was used to classify samples instead of S. Samples below the S/RBD cutoff were classified as naïve, irrespective of the N IgG result.

In Lab B, SARS-CoV-2 and control antigens listed in Table S1 are covalently coupled to Magplex-C microspheres ($1 \times 10^6$ per mL), as described (1). Bead mixes were prepared

**TABLE 1** Optimized assay conditions for Lab A and Lab B SARS-CoV-2 MIAs

| Sample type | Lab A | | | Lab B | | |
|---|---|---|---|---|---|---|
| | DBS | Oral fluid | Serum | DBS | Oral fluid | Serum |
| Plate format | 96 well | | | 384 well, nonbinding (Greiner Bio-One, Monroe, NC) | | |
| Instrument | Luminex MAGPIX | | | Luminex FLEXMAP 3D | | |
| Sample dilution | 1:1500 | Neat | 1:200 | 1:138 | Neat | 1:101 |
| Assay buffer | PBS-TBN | | | PBS + 2% BSA | PBS-TBN | |
| Sample and bead volume (µL) | 12.5, 37.5 | 10, 40 | | 25, 25 | 10, 40 | 25, 25 |
| Incubation conditions | 1 hr. @ RT, shaking at 500 rpm | | | 30 min. @ 37°C, shaking at 300 rpm | | |
| Secondary Ab (concentration) | Phycoerythrin-tagged goat anti-human IgG (5 µg/mL) | | | Phycoerythrin-tagged goat anti-human IgG (4 µg/mL) (ThermoFisher Scientific, Waltham, MA) | | |
| Wash buffer | PBS + 0.05% Tween 20 | | | PBS + 2% BSA + 0.02% Tween 20 + 0.05% sodium azide | PBS + 0.05% Tween 20 | |
| Washer, wash volume (µL) | BioTek 50 TS (Agilent, Santa Clara, CA) 200 | | | BioTek 405 TSUS (Agilent, Santa Clara, CA), 50 | | |
| Resuspension buffer | PBS-TBN | | | xMap sheath fluid (Luminex Corp., Austin, TX) | | |
| Resuspension volume (µL) | 100 | | | 90 | | |
| Resuspension conditions | 1 min. @ RT, shaking at 500 rpm | | | 1 min. @ RT, shaking at 300 rpm with a foil lid | | |
| Minimum bead count | 50 | | | 50 | | |

using 1.25 µL of each bead and 13.75 µL of assay buffer (1,250 beads/bead set/well) for cDBS and serum. For cOF, the bead mix was prepared using 2 µL of each bead and 22 µL of assay buffer (2,000 beads/bead set/well). cDBS were eluted and tested as described (4). Sera were diluted 1:101 using PBS-TBN in a round-bottom, nontreated polystyrene 96-well plate (Corning, Corning, NY). cOF (neat) were aliquoted into a round-bottom, nontreated polystyrene 96-well plate (Corning). Samples and beads are incubated following the protocol outlined in Table 1. Samples were washed twice, incubated with a secondary antibody, and washed twice more (Table 1). Samples were resuspended (Table 1) and analyzed using a FLEXMAP 3D instrument (Luminex Corp., Austin, TX). Results are reported as median fluorescent intensity (MFI) for each bead set. Cutoff values were set for each antigen based on the analysis of 87 pre-COVID pandemic sera (Access Biologicals, Vista, CA). The reactive cutoff was mean MFI+6 standard deviations for S and N antigens for cDBS and S antigens for cOF. The reactive cutoff was mean MFI+3 standard deviations for N antigens in serum and cOF (Table S2). N antigens have a higher level of nonspecific binding in naïve sera and cOF, which increases the standard deviation and results in an artificially inflated reactive cutoff when the +6 standard deviation cutoff is used (Fig. S1). Specimens reactive for ≥2 of the four spike antigens (RBD, S1, FLS, and TRI) are classified as Spike reactive. Index values were calculated by dividing the sample MFI by the reactive cutoff for each bead set. Index values and $\log_{10}$ index values were used for plotting. An index value >1, equivalent to a log index >0, is considered reactive.

## Preparation of contrived DBS and OF

Contrived specimens were prepared using commercially available serum panels and one confirmed Delta-positive specimen (Table S3). Freshly collected Type O EDTA whole blood (ZenBio Inc., Durham, NC) was centrifuged for 8 minutes at 2,200 RCF, and plasma was removed. Because most commercially available whole blood now contains SARS-CoV-2 antibodies, the blood cells were transferred to a sterile tube and triple-washed by performing the following three times: adding an equal volume of PBS as cells, mixing by

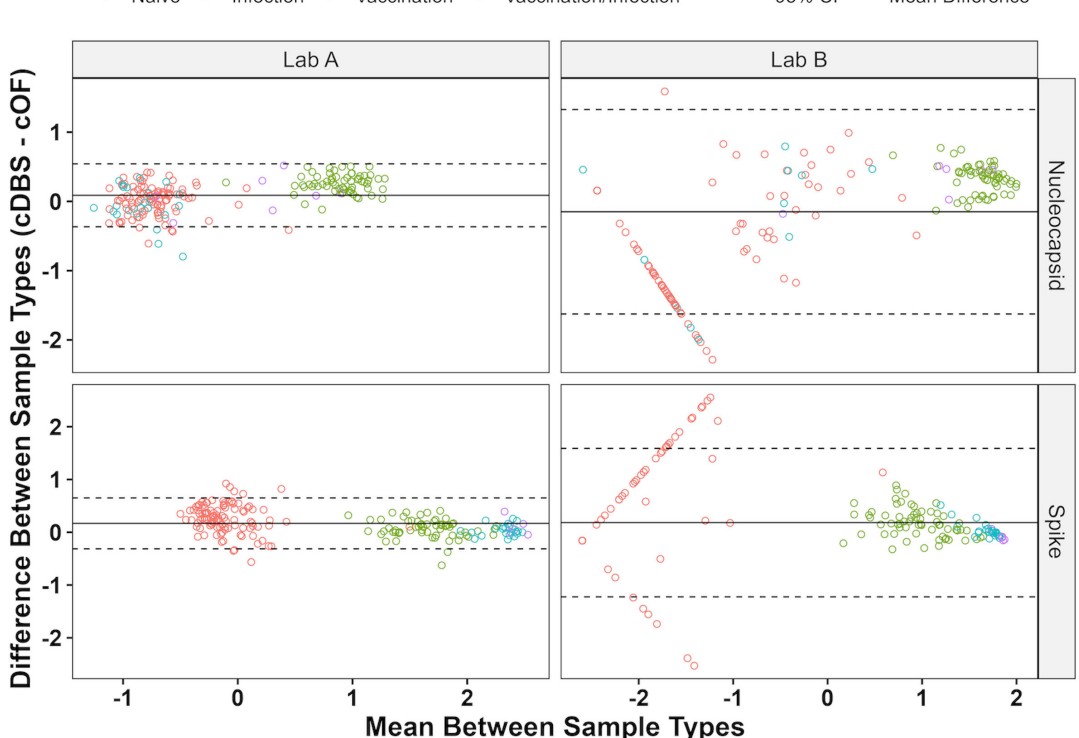

**FIG 1** Intra-laboratory Bland–Altman analysis for Labs A and B assays. Bland–Altman plot of 197 paired samples, with comparisons made between cDBS and cOF for Labs A and B MIAs. Known SARS-CoV-2 antibody status is shown.

inversion for 10 minutes, and centrifuging for 8 minutes at 2,200 RCF. cDBS were then prepared as previously described (1). cOF were prepared using Utak SMx Oral Fluid (Utak, Valencia, CA), diluted 1:8 with PBS-TBN. Serum was diluted 1:200 in 1:8 SMx Oral Fluid.

## Antibody determinations in cDBS and cOF

Specimens were tested following the above-described procedures by Labs A and B (Table 1). In addition, Labs A and B exchanged methods and reagents and tested a subset ($n = 52$) of the contrived specimens. There are two aspects of Labs A and B MIAs that are the same: both use Luminex instrumentation, and both test plasma/serum (Table 1). Aside from these components, the two assays differ in their plate format, sample dilutions, bead mix-to-sample ratios, and buffers (Table 1). Method transfer was performed by both laboratories with minimal changes from the originating laboratory's protocol; however, both laboratories kept their respective wash volumes and plate formats while performing the other laboratory's protocol. In addition, Lab B performed Lab A's assay using Luminex FLEXMAP 3D instrumentation due to a lack of access to a Luminex MAGPIX. Lab A did change Luminex instruments to a FLEXMAP 3D while running Lab B's assay to match its protocol.

## Data analysis

For Lab A, index and log index values for Gen N, Mt. Sinai S, and Sino RBD were used for plotting. An index value >1, or a log index value >0, is reactive. BSA-subtracted net MFIs and cutoffs were adjusted for log transformation by replacing negative values with 1. Lab B's reactivity was determined for the five SARS-CoV-2 antigens (N NA, RBD, S1, FLS, and TRI) using the above-described methods. N and NHT were excluded from the analysis due to poor performance in serum and cOF (Fig. S1).

Qualitative intra- and interlaboratory concordance was assessed using percent agreement, Cohen's kappa, and Fleiss' kappa, as calculated using R package irr (v0.84.1; Gamer et al., 2019). Quantitative intra- and interlaboratory concordance was assessed using Bland–Altman plots, with a threshold of 95% of data points falling within the 95% CI to be considered concordant (8). All analyses and figures were performed in R (v4.3.0; R Core Team, 2023). All figures were created using ggplot2 (v3.4.2; Wickham, 2016) and ggpubr (v0.6.0; Kassambara, 2023). Specimen reactivity from the method transfer between Labs A and B was assigned as described above.

TABLE 2   Intra-laboratory kappa coefficient and percent agreement between sample types and expected results for Labs A and B MIAs (*$P = 0$)[a]

| Laboratory | Sample types compared | Overall reactivity | | | |
| | | Nucleocapsid | | Spike | |
| | | $\kappa$* | % | $\kappa$* | % |
|---|---|---|---|---|---|
| Lab A | cDBS & cOF | 1.00 | 100.0 | 1.00 | 100.0 |
| | cDBS, cOF, & serum[b] | 0.99 | 99.5 | 0.99 | 99.5 |
| | cDBS, cOF, serum, & expected[b] | 0.97 | 97.5 | 0.99 | 99.0 |
| Lab B | cDBS & cOF | 0.97 | 98.5 | 0.98 | 99.0 |
| | cDBS, cOF, & serum[b] | 0.98 | 98.5 | 0.98 | 98.5 |
| | cDBS, cOF, serum, & expected[b] | 0.96 | 97.0 | 0.98 | 98.0 |

[a]Kappa judgment: 0.81–0.99, almost perfect agreement; 1, perfect agreement.
[b]Fleiss' kappa.

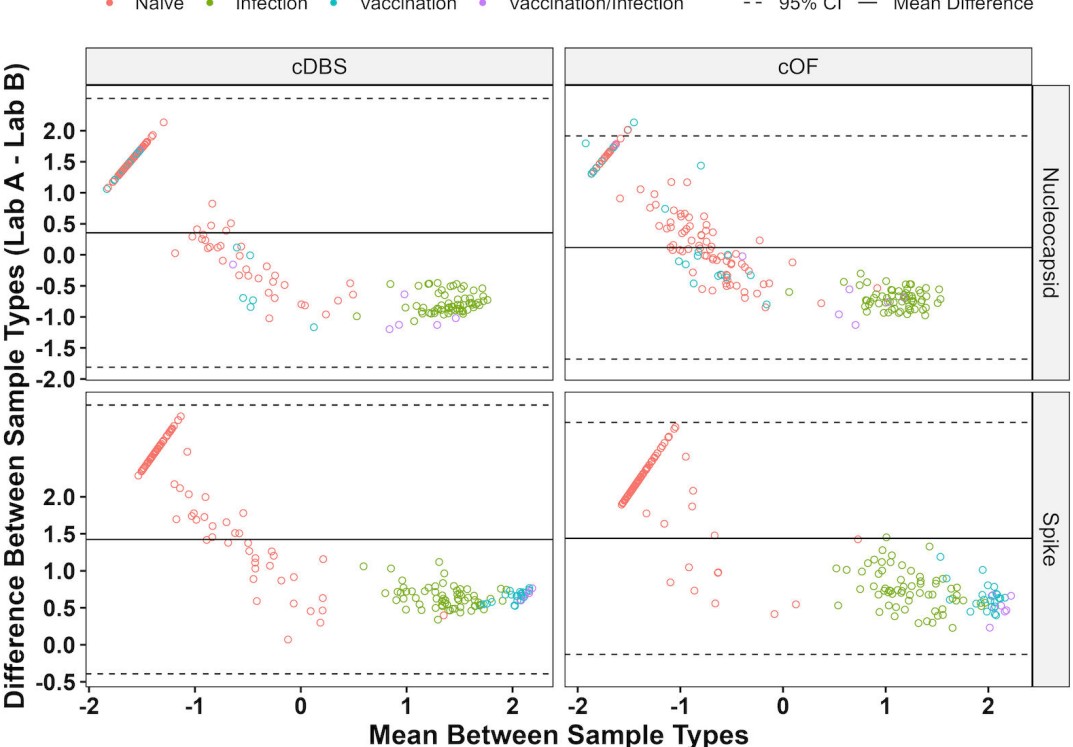

**FIG 2** Interlaboratory Bland–Altman analysis for Labs A and B assays. Bland–Altman plot of 197 paired cDBS and cOF, with comparisons made between Labs A and B performing their respective MIAs. Lab A's MIA uses Mt. Sinai S for cDBS and Sino RBD for cOF, while Lab B's MIA uses four spike antigens (RBD, S1, FLS, and TRI). Known SARS-CoV-2 antibody status is shown.

## RESULTS AND DISCUSSION

### Intra-laboratory cDBS and cOF concordance

To assess intra-laboratory concordance between sample types, a total of 197 serum samples were used to generate paired cDBS and cOF (Table S3). First, we assessed the within-laboratory concordance between cDBS and cOF for Lab A's MIA. Kappa coefficients of 1 (100% agreement) for both N and S reactivities in this MIA were achieved, indicating perfect qualitative agreement between cDBS and cOF (Table 2). N and S reactivities also aligned with the known SARS-CoV-2 status (κ = 0.97–0.99; 97.5%–99% agreement) of donor serum samples. Thus, qualitative concordance between cDBS and cOF was achieved using Lab A's MIA. Quantitative concordance was also achieved for N- and S-reactive samples based on Bland–Altman analysis (8) (100.0% and 97.9% within the 95% CI, respectively) (Fig. 1; Table S4).

Lab B also achieved qualitative and quantitative concordance between cDBS and cOF, but two modifications to the original 8-plex MIA were required. First, the BSA-containing buffers used in that MIA proved to be incompatible with cOF. As a result, Lab B adopted Lab A's buffers for testing cOF specimens (Table S5). Second, the N and NHT antigens in the 8-plex MIA exhibited an unusually high level of nonspecific binding in cOF, resulting in elevated cutoff values and false-negative classifications (Fig. S1). The removal of the N and NHT antigens improved assay sensitivity (Fig. S1). Consequently, qualitative concordance between cDBS and cOF in Lab B's MIA assay was high, based on kappa coefficients for N (κ = 0.97, 98.5% agreement) and S (κ = 0.98, 99% agreement) (Table 2). N and S reactivities also agreed with known SARS-CoV-2 status (κ = 0.96 and 0.98; 97% and 98% agreement, respectively).

Bland–Altman analysis (8) indicates that Lab B's assay produces quantitatively concordant results for both N- and S-reactive cDBS and cOF (both 100% within 95% CI) (Fig. 1; Table S4).

## Interlaboratory concordance of cDBS and cOF

Next, we assessed qualitative concordance between Labs A and B MIA results for cDBS and cOF. Kappa coefficients indicate perfect qualitative agreement for N and S in both sample types (κ = 1.0), with >98% agreement between Lab A and Lab B's cDBS and cOF results. Moreover, those classifications matched the known SARS-CoV-2 status (κ = 0.96–0.99; 95.4%–98.5% agreement) (Table 3). Furthermore, cDBS- and cOF-positive samples were quantitatively concordant for N and S antigens in 100.0% of positive samples within the 95% CI), as determined by Bland–Altman analysis (Fig. 2; Table S6). In conclusion, Labs A and B successfully demonstrated interlaboratory qualitative and quantitative concordance for cDBS and cOF.

## Method transfer

For the MIA assay platform to be useful within the public health arena, the assays must be transferable between laboratories. To address the issue of transferability, Lab A adopted Lab B's assay, and Lab B adopted Lab A's assay using a total of 52 paired OF and DBS samples. When Lab A's assay was performed by labs A and B (Lab A in A/B), results were qualitatively concordant for both cDBS and cOF (κ ≥ 0.92; ≥98.1% agreement) (Table 3). Kappa coefficients were slightly improved with Lab A's MIA, for reasons that are unclear (Table 3). Additionally, Bland–Altman analysis showed that the 52 paired samples were quantitatively concordant (100% within 95% CI) for SARS-CoV-2-reactive cDBS and cOF samples with lab A's assay (Lab A in A/B) (Fig. 3; Table S6)

Similarly, when the Lab B assay was performed by labs A and B (Lab B in A/B), results were qualitatively and quantitatively concordant for N (κ = 0.61–1, 82.7%–100% agreement) and S (κ = 0.96–1, 96.2%–100% agreement) (Table 3). Bland–Altman analysis showed that the 52 paired samples were quantitatively concordant for Lab B's MIA

**TABLE 3** Kappa coefficient and percent agreement across sample types and expected results for Labs A and B MIAs (*P = 0)[a]

| Lab(s) | MIA method performed | Sample types compared | Overall reactivity | | | |
|---|---|---|---|---|---|---|
| | | | Nucleocapsid | | Spike | |
| | | | κ* | % | κ* | % |
| Labs A and B | Labs A and B, respectively | cDBS | 1.0 | 98.0 | 1.0 | 99.0 |
| | | Serum | 0.94 | 97.5 | 0.99 | 99.5 |
| | | cOF | 0.94 | 97.5 | 1.0 | 100.0 |
| | | cDBS, cOF, serum[b] | 0.96 | 96.5 | 0.99 | 98.5 |
| | | cDBS, cOF, serum, expected[b] | 0.96 | 95.4 | 0.99 | 98.0 |
| Labs A and B | Lab A | cDBS | 1.0 | 100.0 | 0.96 | 98.1 |
| | | Serum | 1.0 | 100.0 | 0.96 | 98.1 |
| | | cOF | 0.96 | 98.1 | 0.92 | 96.2 |
| | | cDBS, cOF, serum[b] | 0.99 | 98.1 | 0.96 | 96.2 |
| Labs A and B | Lab B | cDBS | 1.0 | 100.0 | 1.0 | 100.0 |
| | | Serum | 0.92 | 96.2 | 0.96 | 98.1 |
| | | cOF | 0.61 | 82.7 | 0.96 | 98.1 |
| | | cDBS, cOF, serum[b] | 0.86 | 82.7 | 0.97 | 96.2 |

[a]Kappa judgment: 0.81–0.99, almost perfect agreement; 1, perfect agreement.
[b]Fleiss' kappa.

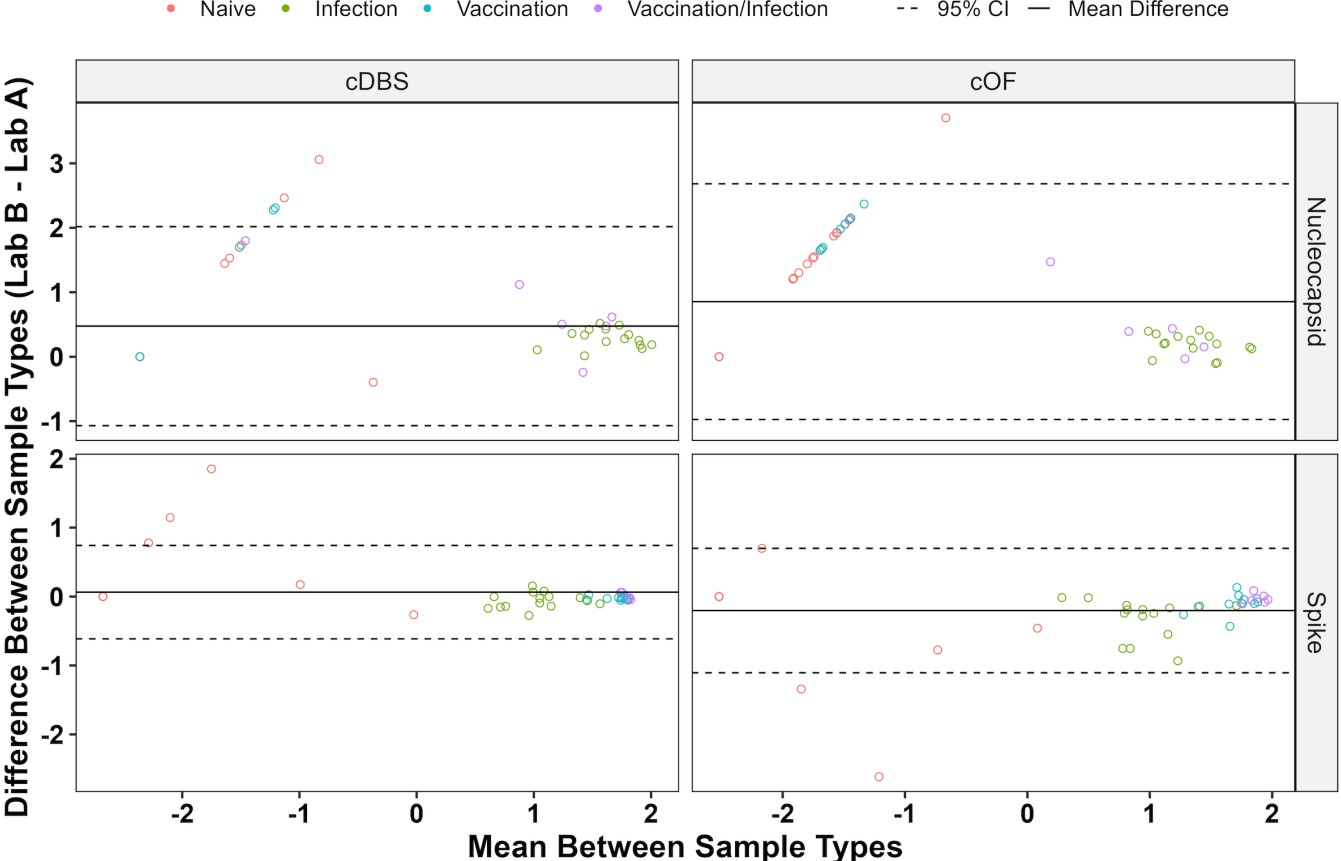

**FIG 3** Interlaboratory Bland–Altman analysis for the Lab A assay. Bland–Altman plot of a subset of samples (*n* = 52) cDBS and cOF tested by Labs A and B using Lab A's MIA. Known SARS-CoV-2 antibody status is shown. An interlaboratory comparison of the Lab B assay is shown in Fig. S2.

with >96.9% of cDBS- and cOF SARS-CoV-2-positive samples falling within the 95% CI (Table S6; Fig. S2). In conclusion, labs A and B obtained concordant qualitative and quantitative interlaboratory results for cDBS and cOF when performing method transfer.

## Conclusions and study limitations

In this report, we investigated the congruity between DBS and OF as model biospecimen types for use in SARS-CoV-2 serological studies. Contrived DBS (cDBS) and contrived OF (cOF) from serum panels of SARS-CoV-2-infected, vaccinated, and uninfected individuals were generated and evaluated using two different SARS-CoV-2 MIAs at two different institutions. With some minor modifications of MIA protocols, we found that DBS and OF were largely interchangeable across laboratories for the qualitative assessment of SARS-CoV-2 antibody determinations. Qualitative assessment was achieved using the same instrument platform (Luminex), but different antigen panels, wash buffers, and internal controls demonstrate that the biospecimens themselves are sufficiently robust to enable interinstitutional comparisons. An obvious limitation of this study is the reliance on contrived biospecimens (DBS and OF), thereby avoiding the inherent variability associated with samples in the field. OF is not a homogenous sample, and numerous factors can impact antibody concentration and sample viscosity (3, 9). Additionally, salivary antibody levels are known to vary from person to person and throughout the day in individuals. While DBS are known to be temperature-stable for shipping, little is known about OF stability during transportation (10–12). Nonetheless, the methods developed in this study are being tested using authentic paired DBS and OF samples to demonstrate intra- and interlaboratory concordance of these two sample types. Future efforts will assess IgG and IgA concordance of SARS-CoV-2 antibodies

between paired DBS and OF, the applicability of this approach to other detection platforms, and other factors (e.g., operator usability) that may influence concordance. Indeed, it is notable that DBS and oral mucosal transudate (OMT) have been successfully demonstrated to discriminate between HPV-vaccinated and unvaccinated individuals based solely on antibody titers to two HPV antigens (13).

## ACKNOWLEDGMENTS

We thank Drs. Monica Parker and William Lee of the Wadsworth Center for supervision and technical assistance. We gratefully acknowledge Elizabeth Cavosie for administrative assistance.

This study was supported by the National Cancer Institute (NCI) of the National Institutes of Health (NIH) under award number U01CA260508. The content is solely the responsibility of the authors and does not necessarily represent the official views of the NIH.

## AUTHOR AFFILIATIONS

[1]New York State Department of Health, Wadsworth Center, Albany, New York, USA
[2]Department of Environmental Health and Engineering, Bloomberg School of Public Health, Johns Hopkins University, Baltimore, Maryland, USA
[3]Department of Epidemiology, Bloomberg School of Public Health, Johns Hopkins University, Baltimore, Maryland, USA
[4]Department of International Health, Bloomberg School of Public Health, Johns Hopkins University, Baltimore, Maryland, USA
[5]Biomedical Sciences Department, State University of New York at Albany College of Integrated Health Sciences, Albany, New York, USA

## AUTHOR ORCIDs

Kate L. DeRosa http://orcid.org/0000-0003-0280-2667
Christopher D. Heaney http://orcid.org/0000-0003-3211-8495
Linda M. Styer http://orcid.org/0000-0001-8137-0318
Nicholas J. Mantis http://orcid.org/0000-0002-5083-8640

## FUNDING

| Funder | Grant(s) | Author(s) |
| --- | --- | --- |
| National Cancer Institute | U01CA260508 | Nicholas J. Mantis |

## AUTHOR CONTRIBUTIONS

Kate L. DeRosa, Formal analysis, Investigation, Methodology, Writing – original draft, Writing – review and editing | Nora Pisanic, Formal analysis, Investigation, Methodology, Resources, Writing – original draft, Writing – review and editing | Kate Kruczynski, Investigation, Methodology | Christopher D. Heaney, Conceptualization, Formal analysis, Funding acquisition, Investigation, Methodology, Project administration, Writing – original draft, Writing – review and editing | Linda M. Styer, Conceptualization, Formal analysis, Investigation, Methodology, Project administration, Supervision, Writing – original draft, Writing – review and editing | Nicholas J. Mantis, Conceptualization, Funding acquisition, Investigation, Project administration, Supervision, Writing – original draft, Writing – review and editing

## DATA AVAILABILITY

The raw MFI values obtained from Labs A and B for each set of experiments described in this study are provided as Datafile S1.

## ADDITIONAL FILES

The following material is available online.

## Supplemental Material

**Datafile S1 (Spectrum02690-24-s0001.xlsx).** The raw MFI values obtained from Lab A and Lab B for each set of experiments described in this study.
**Supplemental tables and figures (Spectrum02690-24-s0002.pdf).** Tables S1 to S6; Figures S1 and S2.

## Open Peer Review

**PEER REVIEW HISTORY (review-history.pdf).** An accounting of the reviewer comments and feedback.

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
