## [Reviewer comments · Microbiology Spectrum]

Microbiology Spectrum

Inter-laboratory harmonization of microsphere immunoassays for SARS-CoV-2 antibody detection in contrived dried blood spots and oral fluids

Kate DeRosa, Nora Pisanic, Kate Kruczynski, Christopher Heaney, Linda Styer, and Nicholas Mantis

Corresponding Author(s): Nicholas Mantis, Wadsworth Center, New York State Department of Health

Review Timeline:

Submission Date:	October 23, 2024
Editorial Decision:	January 8, 2025
Revision Received:	January 30, 2025
Accepted:	February 7, 2025

Editor: Alex Dulovic

Reviewer(s): Disclosure of reviewer identity is with reference to reviewer comments included in decision letter(s). The following individuals involved in review of your submission have agreed to reveal their identity: Dean D Erdman (Reviewer #2)

Transaction Report:

DOI: <https://doi.org/10.1128/spectrum.02690-24>

Re: Spectrum02690-24 (Inter-laboratory harmonization of microsphere immunoassays for SARS-CoV-2 antibody detection in dried blood spots and oral fluids)

Dear Dr. Nicholas J. Mantis:

Thank you for the privilege of reviewing your work. Below you will find my comments, instructions from the Spectrum editorial office, and the reviewer comments.

In response to the comments raised by the reviewers, please consider including the word "contrived" as part of the title.

Revision Guidelines

Sincerely,
Alex Dulovic
Editor
Microbiology Spectrum

Reviewer #1 (Comments for the Author):

The authors present their work using a multiplexed immunoassay to evaluate DBS and OF specimen types for detection of SARS-CoV-2 antibodies to determine interchangeability of the specimen types and also the ability to achieve consistent results between different laboratories.

The authors have shown that these assays perform well and are robust to different antigen sources, different buffer systems, different assay protocols, different instrumentation, and different methods for data analysis and establishing cutoffs.

Using data normalization is key since the different instruments generate different raw signals but can show identical/near identical results when normalized.

The authors acknowledge the limitations due to using contrived specimens here but the data are sufficient to support further work with larger sets of authentic paired samples and more samples for inter-laboratory comparison. They might also consider sample exchange with additional laboratories using similar methods on this technology.

Have the results been compared to results generated on other/different technologies?

Another thought is if a similar study has been done for pathogens other than SARS-CoV-2? Are these results unique for this virus or are similar results seen for immunoassays for other pathogens?

Reviewer #2 (Comments for the Author):

Journal: Microbiology Spectrum

Type of manuscript: Research Article

Manuscript ID: Spectrum02690-24

Title: Inter-laboratory harmonization of microsphere immunoassays for SARS-CoV-2 antibody detection in dried blood spots and oral fluids

Summary and general comments: This study compares test results obtained for detection of SARS-CoV-2 antibodies in contrived dried blood spot and oral fluid specimens using microsphere immunoassay (Luminex) technology within and between laboratories, designated A and B. It furthers the comparison by swapping assays between laboratories. Overall agreements between the sample types and testing sites were high.

Specific comments:

- 1) Title. The major study limitation acknowledged by the authors is their use of constructed or "contrived" specimens (as compared with fresh authentic clinical samples) for assay comparisons. As such, "contrived" should appear somewhere in the manuscript title.
- 2) Materials & Methods. Line 71. Spell out "MIA" when first used in the body of the text.
- 3) Materials & Methods. If more than just convenience, what criteria were used to select laboratories A and B for test comparisons? It is concerning that laboratory B's assay needed modification and data on some antigens were excluded to obtain comparable results with laboratory A.
- 4) Materials & Methods. What were the methods used to determine how the commercial sera were originally classified given these represent the Gold Standard comparator?
- 5) Materials & Methods. Did the same technicians perform all testing for this study?
- 6) Results & Figure S2. Were outlier samples retested?
- 7) Table S2. Units should be provided.
- 8) Table S6. Typos "100.00".

Response to Reviewers for Spectrum02690-24

Reviewer #1 (Comments for the Author):

The authors present their work using a multiplexed immunoassay to evaluate DBS and OF specimen types for detection of SARS-CoV-2 antibodies to determine interchangeability of the specimen types and also the ability to achieve consistent results between different laboratories. The authors have shown that these assays perform well and are robust to different antigen sources, different buffer systems, different assay protocols, different instrumentation, and different methods for data analysis and establishing cutoffs. Using data normalization is key since the different instruments generate different raw signals but can show identical/near identical results when normalized.

R1.1 The authors acknowledge the limitations due to using contrived specimens here but the data are sufficient to support further work with larger sets of authentic paired samples and more samples for inter-laboratory comparison. They might also consider sample exchange with additional laboratories using similar methods on this technology. Response: Thank you for the suggestion. We will explore this option as we expand the study to include authentic paired samples.

R1.2 Have the results been compared to results generated on other/different technologies? Response: The results have only been compared using Luminex. We note this as a limitation of the study in the conclusions section.

R1.3 Another thought is if a similar study has been done for pathogens other than SARS-CoV-2? Are these results unique for this virus or are similar results seen for immunoassays for other pathogens? Response: Excellent point. We have updated the conclusions section to point out the work by Louie and colleagues demonstrating “Indeed it is notable that it has been successfully demonstrated that DBS and oral mucosal transudate (OMT) could each be used to discriminate between HPV vaccinated and unvaccinated individuals based solely on antibody titers to two HPV antigens¹³. [Citation: Louie KS, Dalel J, Reuter C, Bissett SL, Kleeman M, Ashdown-Barr L, Banwait R, Godi A, Sasieni P, Beddows S. Evaluation of Dried Blood Spots and Oral Fluids as Alternatives to Serum for Human Papillomavirus Antibody Surveillance. mSphere. 2018 May 9;3(3):e00043-18. doi: 10.1128/mSphere.00043-18. PMID: 29743199; PMCID: PMC5956145.]

Reviewer #2 (Comments for the Author):

Summary and general comments: This study compares test results obtained for detection of SARS-CoV-2 antibodies in contrived dried blood spot and oral fluid specimens using microsphere immunoassay (Luminex) technology within and between laboratories, designated A and B. It furthers the comparison by swapping assays between laboratories. Overall agreements between the sample types and testing sites were high.

Specific comments:

R2.1 Title. The major study limitation acknowledged by the authors is their use of constructed or "contrived" specimens (as compared with fresh authentic clinical samples) for assay comparisons. As such, "contrived" should appear somewhere in the manuscript title. Response: As requested, we have inserted "contrived" into the manuscript title.

R2.2 Materials & Methods. Line 71. Spell out "MIA" when first used in the body of the text. Response: Updated, as per ASM guidelines.

R2.3 Materials & Methods. If more than just convenience, what criteria were used to select laboratories A and B for test comparisons? It is concerning that laboratory B's assay needed modification and data on some antigens were excluded to obtain comparable results with laboratory A. Response: Laboratories A and B were selected based on their respective expertise in the analysis of oral fluids (Lab A) and DBS (Lab B) plus their shared use of the Luminex platform. Therefore, it was a strategic partnership more than convenience.

R2.4 Materials & Methods. What were the methods used to determine how the commercial sera were originally classified given these represent the Gold Standard comparator? Response: Panel D sera were collected 2-3 months after individuals had tested positive for SARS-CoV-2 infection or received a medical diagnosis of SARS-CoV-2. All sera in this panel also tested positive on the DiaSorin Liaison SARS-CoV-2 S1/S2 IgG antibody assay and the Gold Standard Diagnostics SARS-CoV-2 IgG assay. Panel E sera were collected in 2017 and 2018, prior to the emergence of SARS-CoV-2 and were expected to be negative. Panel F sera were collected 1-4 months after individuals were diagnosed with SARS-CoV-2; all sera tested positive on the DiaSorin Liaison SARS-CoV-2 S1/S2 IgG antibody assay and the Centaur SARS-CoV-2 antibody assay. Panel H sera were tested using the DiaSorin Liaison assay and we have used the results of this assay to determine expected results. The expected results of the seroconversion panel were based on the results of the DiaSorin Liaison SARS-CoV-2 S1/S2 IgG assay, Gold Standard Diagnostics SARS-CoV-2 IgG assay, and the Ortho Vitros Anti-SARS-CoV-2 IgG and Anti-SARS-CoV-2 Total tests. Other commercial samples were obtained from individuals known to be infected with SARS-CoV-2 but antibody results were not provided with the samples. We confirmed the positive antibody result with our own assay.

R2.5 Materials & Methods. Did the same technicians perform all testing for this study? Response: The same technicians in laboratory A and B performed the testing. As such, the study did not examine inter-operator variability. as the focus was specimen type and inter-laboratory harmonization. This is a limitation of the study that we now underscore in the study limitations section.

R2.6 Results & Figure S2. Were outlier samples retested? Response: All specimens were run in triplicate. If the same “outliers” occurred in the replicates the samples were not re-tested.

R2.7 Table S2. Units should be provided. Response: Table S2 has been updated to spell out MFI – median fluorescence intensity, which is the unit of the cutoff values.

R2.8 Table S6. Typos "100.00". Response: Corrected.

Re: Spectrum02690-24R1 (Inter-laboratory harmonization of microsphere immunoassays for SARS-CoV-2 antibody detection in contrived dried blood spots and oral fluids)

Dear Dr. Nicholas J. Mantis:

Thank you for submitting a revised version of your manuscript. I am pleased to inform you that your manuscript has been accepted, and I am forwarding it to the ASM production staff for publication. Your paper will first be checked to make sure all elements meet the technical requirements. ASM staff will contact you if anything needs to be revised before copyediting and production can begin. Otherwise, you will be notified when your proofs are ready to be viewed.

Sincerely,
Alex Dulovic
Editor
Microbiology Spectrum